# Stress across the Lifespan: From Risk to Management—Conference Report on the Inaugural Canadian Stress Research Summit

**DOI:** 10.3390/ijerph191711015

**Published:** 2022-09-03

**Authors:** Alexandra J. Fiocco

**Affiliations:** Institute for Stress and Wellbeing Research, Department of Psychology, Toronto Metropolitan University, 350 Victoria St., Toronto, ON M5B 2K3, Canada; afiocco@ryerson.ca

**Keywords:** stress, research, knowledge translation, Canada

## Abstract

In 2021, the Toronto Metropolitan University Institute for Stress and Wellbeing Research welcomed over 200 conference delegates across Canada to the inaugural Canadian Stress Research Summit (CSRS) to share ideas and foster collaboration among Canadian scholars. This conference was unique from existing international stress-related conferences as it bridged science and community. The objective of this conference report is to provide an overview of the 3-day virtual inaugural stress conference, offering a summary of the keynote addresses, themed symposia, spotlight presentations, graphical designs of selected presentations, and conference feedback. Overall, the CSRS highlighted important methodological considerations in understanding the relationship between stress exposure and various outcomes of interest that pertain to the mental health and wellbeing of Canadians. Furthermore, there is a need for continued work to understand stress across the lifespan from an inclusive and diverse Canadian lens.

## 1. Introduction

Chronic perceived stress is an insidious phenomenon associated with adverse health outcomes. Research suggests that chronic or intermittent activation of stress-sensitive systems is associated with increased risk for various neurological, psychological, and physical ailments across the lifespan [1,2]. In Canada, over 1 in 4 Canadians report daily stress levels in the high to severe range [3]. Stress-related mental health ailments cost employers approximately CAD 20 billion dollars a year and account for over three-quarters of short-term disability claims in Canada [4]. Consequently, the development of low-cost strategies that minimize the effect of stress on individuals of all ages is paramount. It is therefore important to bring together scientists, clinicians, and innovators who are dedicated to turning stress-related theory into practice.

In May 2021, the inaugural Canadian Stress Research Summit (CSRS) was held virtually by Toronto Metropolitan University’s Institute for Stress and Wellbeing Research. The CSRS was the first Canadian conference of its kind, dedicated to the dissemination of stress research in Canada. The theme of the inaugural meeting was Stress Across the Lifespan: From Risk to Management. This scientific conference hosted over 200 delegates including scientists, clinicians, trainees, and community members. The goal of this 3-day conference was to engage stakeholder groups and to foster cross-pollination of ideas and collaborations to improve our understanding of stress physiology, its mechanisms, behavioral outcomes, and mitigation strategies through a diverse Canadian lens. This conference was unique from existing international stress-related conferences as it bridged science and community. 

The objective of this conference report is to provide an overview of the 2021 CSRS, offering a summary of the keynote addresses, themed symposia, and spotlight presentations. Graphical representations synthesizing selected presentations and scientific abstracts are displayed in the Appendix A.

## 2. Conference Inception and Intentions

Canada has a rich history of scholars who have dedicated their programs of research to understanding stress and elucidating the mechanisms of stress-related pathology. Indeed, the pioneering work of Hans Selye, who coined to term “stress”, revolutionized our understanding of the causes and mechanisms of disease, advancing the theory of general adaptation syndrome, which recognizes that stress plays a crucial role in disease development [5]. Given the unique history of stress research in Canada, it was surprising that a national stress conference was not available for Canadian stress researchers to foster networking, collaboration, and the synergistic training of future scholars.

In 2012, the Department of Psychology at Toronto Metropolitan University founded the Institute for Stress and Wellbeing Research. The inception of a national conference was developed through conversations among members of the Institute’s Governance Committee, which has a mandate to promote quality of life and health among Canadians through the creation of synergistic research collaborations that advance knowledge and practical applications in the areas of stress, wellbeing, intervention, and policy.

In 2018, a conference planning committee was created that included one member of the Institute’s Governance Committee (Alexandra J Fiocco), the Department of Psychology Research Operations Administer (Carson Pun), and five graduate students (Jessica Burdo, Danielle D’Amico, Rachel Goren, Brittany Jamieson, Shruti Vyas, and Sally Zheng). Together, the conference planning committee shaped the intentions of the conference and developed the conference name and theme of the inaugural meeting. As the first scientific meeting of the CSRS, it was agreed that a broad theme should be implemented, one that encompassed a lifespan approach from basic science to intervention-based human research.

The intention of the conference was to provide a unique platform for Canadian scholars in the field of stress research, one that celebrates the rich history of stress research in Canada and highlights emerging Canadian research in the field. Given the Institute’s emphasis on practical application and the wellbeing of Canadians, the planning committee deemed it important to offer accessible knowledge dissemination for the lay community. As such, in addition to the creation of scientific symposia and poster presentation sessions, the committee embedded lay community presentations and workshops into the conference agenda. Furthermore, graphic recordings were implemented in order to promote engagement and better understanding of the presented material. While the initial intention was to offer the inaugural meeting in-person at Toronto Metropolitan University, the COVID-19 pandemic resulted in a necessary pivot to offering a virtual meeting.

After securing two keynote speakers, lay community presenters, and workshop facilitators, a call for abstracts was circulated, requesting scientific abstracts that fell within the broader context of stress across the lifespan. Suggested topics included stress biomarkers and mechanisms; stress interventions and resilience; stress, sex, and gender; stress and Indigenous health; stress and discrimination; stress and development; stress and aging; stress and clinical outcomes; and stress and COVID-19. Abstract submissions were in the form of poster presentations, student flash-five talks, or oral presentations. The conference planning committee reviewed abstract submissions and created the themed symposium sessions.

## 3. Conference Structure

In response to the COVID-19 pandemic, the inaugural conference entailed a 3-day virtual program using a conferencing platform called EventMobi. Table 1 displays an overview of the program. Presenters were required to provide a prerecording of their presentation for each symposium session, except for the Indigenous Health Symposium and the General Public Presentation. 

The CSRS Poster Session was held using Gather, a customizable virtual meeting space, to facilitate an interactive and engaging poster session. All posters were received before the conference date and were uploaded onto Gather’s virtual poster boards by the conference committee. A roof top lounge was created to allow conference delegates to meet outside of the symposium sessions. A snapshot of the poster room and roof top lounge are displayed in Appendix A. 

## 4. Conference Sessions

### 4.1. Keynote Presentations

The first day of the scientific conference welcomed Sonia Lupien, a Professor at the Université de Montréal (Montreal, Quebec) and Founder and Director of the Centre for Studies on Human Stress [6]. Lupien’s keynote presentation, entitled “From Neurotoxicity to Vulnerability: A Developmental Perspective of the Effect of Stress on the Brain”, encapsulated the theme of the 2021 conference, providing an overview of 30 years of research demonstrating the profound impact that chronic stress and glucocorticoid secretion can have on the brain at different developmental stages and the importance of intervention strategies to enhance resilience among Canadians. As the opening keynote speaker, Lupien’s presentation catered to all conference delegates including senior scientists, trainees, clinicians, and community members alike. A graphical synthesis of Lupien’s keynote presentation is displayed in Appendix A. 

Matthew Hill, a Professor at the University of Calgary (Calgary, Alberta), provided the second conference keynote presentation, entitled “A Tale of Translation: Endocannabinoid Regulation of Stress, Anxiety, and Fear, From Rodent to Humans” [7]. Hill provided an overview of their research on the role of tonic endocannabinoid signaling in the regulation of stress-like states. Bridging preclinical animal research and human genetic and pharmacological data, Hill presented a streamlined translation of the role of anandamide (AEA) and fatty acid amide hydrolase (FAAH) in regulating neural activity within the basolateral amygdala and the ensuing physiological and behavioral stress response. A graphical synthesis of Hill’s keynote presentation is displayed in Appendix A.

### 4.2. Symposia Highlights

One goal of the scientific committee was to include a wide range of topics that reflected the conference’s theme of Stress across the Lifespan: From Risk to Management. Based on the submitted abstracts, a total of eight symposia sessions were created including: Stress, Sex, and Gender; Stress and Indigenous Health; Stress on the Job; Stress Biomarkers and Mechanisms; Stress Interventions and Resilience; Stress and Development; Stress and Clinical Outcomes; Stress and COVID-19. All symposium presentation abstracts are available at https://www.torontomu.ca/content/dam/canadian-srs/CSRS_2021_Abstract_Book.pdf (accessed on 6 May 2021).

The Stress, Sex, and Gender symposium welcomed experts in the field from basic neuroscience to clinical research [8,9,10,11]. Together, presenters highlighted the importance of considering biological sex in order to understand stress physiology, susceptibility, and resilience. The employment of sex-specific stress paradigms (e.g., restraint stress in rodents; [8]), algorithms (e.g., allostatic load calculation; [11]), and statistical modeling (e.g., controlling for sex steroids; [11]) is fundamental to understanding the nuanced mechanisms and outcomes of stress in males and females. Beyond biological sex, sociocultural gender is also important to consider and, in some cases, may underly reported sex differences in the literature [9,10,11]. For example, prenatal stress exposure may result in differential gendered experiences during critical developmental periods when stress-sensitive systems are being calibrated [10]. Presenters highlighted the importance of moving beyond the binary and to examine the intersection between sex, gender, and sexual orientation to better understand stress physiology and stress resilience in the general Canadian population. 

Stress and Indigenous Health: Teachings of Blood Memory and Epigenetics was a special highlighted symposium that was intended to bridge mainstream colonial science and traditional Indigenous teachings, fostering a two-eyed seeing approach on how stress may become biologically embedded through generational trauma among First Nations Peoples in Canada. McQuaid and Bombay [12] provided an overview of their work with First Nations community leaders on the intergenerational effects of residential schools. Traditional Ancestral Knowledge Keeper, Kim Wheatley, is Ojibway and Potawatomi from Shawanaga First Nation, who carries the Spirit name “Head or Leader of the Fireflower”. Wheatley shared the traditional teaching and understanding that blood memory is an intergenerational carrier of memories, teaching, and knowledge of ancestors. This means traumas of the past can also be passed to descendants, affecting their health and wellness by manifesting in a wide variety of body, mind, and spirit imbalances. Wholistic approaches through ceremonial practices focus on restoring balance [13]. A graphical synthesis of this symposium is displayed in Appendix A.

Stress on the Job focused on research conducted with police officers in Canada [14,15,16,17]. When dispatched to a high-priority call, police officers display significant increases in stress physiology in anticipation of arrival at the scene and may further display continued elevations of stress physiology following apprehension of the individual [15]. Heightened stress physiology may compromise de-escalation performance and cognitive processing of events, which may result in occupational stress and poor performance outcomes [16,17]. Automatic modulation training (AMT) and the International Performance, Resilience, and Efficiency Program (iPREP) were presented as potential intervention tools to support police officers in developing the skills needed to mitigate the negative impact of chronic occupational stress on job performance and physical and mental health [14,16]. Implications for policy and practice were presented, which may also be found in a 2021 RSC Policy Briefing [18].

The Stress Biomarkers and Mechanisms symposium featured a range of animal- and human-based research that together provided a biopsychosocial lens to the antecedents and mechanisms of stress [19,20,21,22]. Allostatic load and inflammation were examined as important biomarkers of stress in Canadian firefighters [21] and family caregivers [19], respectively. Presenters spoke of the role of dorsal CA1 hippocampal neurons in differentiating between resilient and stress-susceptible animals [22] and the role of early childhood adversity in altering stress-sensitive systems and decreasing the threshold of activation in response to future stressors [20]. Dispositional characteristics, namely, implicit affect, was also examined as a dispositional characteristic that may associate with automatic appraisals, thus influencing the perception of stress and the stress response that ensues [19].

The Stress Interventions and Resilience symposium featured strength-based approaches for stress management across the lifespan and the recognition that resilience can be conceptualized as a multisystem construct [23,24,25,26]. Presenters provided evidence for school-based programing by the Strong Minds Strong Kids Foundation (Psychology Foundation of Canada) for the cultivation of coping strategies that support emotion regulation in school-aged children [25]. The utility of psychological distancing when reflecting on one’s current stress was also explored [26]. Finally, early evidence from SingWell Canada was presented, illustrating the psychosocial and physiological benefits of group singing among older adults with and without chronic illness [24].

The Stress and Developmental symposium discussed the feasibility of pivoting stress research in children to remote platforms and the role of parental care and childhood adversity on the regulation of stress-sensitive systems [27,28,29,30]. Presenters showed that stress reactivity in young children is not only moderated by the child’s mental health risk status but may also be buffered by parental presence during stress induction [29]. Although current findings are inconclusive [28], parental training interventions to enhance nurturing care have been proposed to help regulate stress-sensitive systems and to foster the wellbeing of the developing child. A sex and gender lens was also incorporated into this symposium, showing that early childhood maltreatment is robustly associated with revictimization among women; however, a more nuanced exploration of the type of maltreatment, the perpetrator (i.e., maternal or paternal), and sexual identity (monosexual vs. plurisexual) is integral to understanding how early childhood maltreatment may result in revictimization later in adulthood [30].

Discussing the important role of early childhood adversity on mental health and wellbeing was extended to the Stress and Clinical Outcomes symposium [31]. However, from a clinical perspective, the importance of taking a cumulative lifespan approach, beyond childhood adversity, was highlighted when evaluating mental health outcomes [32]. Assessment of the type of stressor may also be valuable in determining certain risks within clinical samples. For example, depression among Canadian adolescents is associated with higher stress exposure across multiple domains; however, greater interpersonal stress, in particular, is associated with attempting suicide among adolescents with major depressive disorder [33]. Moving from risk to management, cognitive behavioral conjoint therapy for post-traumatic stress disorder (PTSD) with 3,4 -methylenedioxymethamphetamine (MDMA) was presented as a safe and feasible treatment for decreasing PTSD symptoms and supporting relationship satisfaction [34].

Finally, the conference welcomed early research on the impact of the coronavirus pandemic in Canada. The COVID-19 Pandemic symposium included prospective studies that highlighted individual difference factors that influenced health and wellbeing outcomes during the first year of the pandemic in Canada [35,36,37,38]. For example, in preadolescence, stress-sensitive girls were at greater risk of internalizing symptoms relative to boys during the pandemic [36]. In the context of scholastic performance, prospective evaluation of undergraduate students highlighted the importance of considering stress across the semester to determine final grades and revealed group-based differences such that grades were especially sensitive to COVID-19 stress among females, persons of color, those high in religiosity, and students with a lower income [35]. During the initial pandemic lockdown, relationship dissatisfaction was a significant predictor of engaging in emotional eating one month into home confinement, above and beyond common well-known predictors of emotional eating such as depression and being female [38]. Finally, in examining the later stage of the life course spectrum, the work of Lussier et al. [37], using data from the Translational Biomarkers of Aging and Dementia and TRIAD Assessment of Social Isolation and Cognition project, suggested that persons with greater tau pathology may present with a lower understanding of COVID-19 symptoms and lower levels of pandemic-related stress and disrupted sleep quality, suggesting that public health information may be less accessible to older adults who are further along the spectrum of cognitive decline and dementia. A graphical synthesis of this symposium is displayed in Appendix A.

### 4.3. CSRS Student Spotlight 

A total of nine student presentations were selected for the Student Spotlight session, which consisted of five-minute student presentations followed by a question-and-answer period. The Best Flash-Five prize was awarded to Renda et al. [39] for their presentation entitled “Combined Adolescent Nicotine and Foot Shock Stress Exposure Augments Adult Nicotine Self-Administration without Affecting Adult Baseline Anxiety-like Behavior or Corticosterone Response to Nicotine or Foot Shock Stress”. 

### 4.4. Poster Presentations

The poster session included 50 poster presentations across the eight conference themes. The Best Poster prize was awarded to Arsenault et al. [40] for their poster entitled “Eukaryotic Initiation Factor 4E-Binding Proteins Mediate the Antidepressant Response to Ketamine in Mice Exposed to Chronic Variable Stress”. 

### 4.5. CSRS General Public Presentation and Workshops 

With the intention of bridging science and the lay community, a public presentation on the final day of the conference entitled “Mental Health in 2021: The Echo Pandemic” was delivered by Nasreen Khatri. A graphical synthesis of the public presentation is displayed in Appendix A. Furthermore, two workshops were offered to conference delegates: one intended to support students and trainees, entitled “Establishing a Flourishing Relationship between You and Your Mentor”, by Diana Brecher; the other by Jeremy Finkelstein, entitled “Community Workshop: Mindfulness for Wellbeing”, indented to support conference delegates interested in the utility of mindfulness for stress management. 

## 5. Conference Analytics and Feedback

The conference welcomed a total of 263 conference delegates with an 86.3% login rate. Approximately one week following the conference, a survey was sent out to conference delegates to solicit feedback. Eight percent of the conference delegates completed the post-conference survey. Among the respondents, 31% identified as being an independent researcher, 42% identified as being a student or postdoctoral trainee, 0.05% identified as a clinical or healthcare provider, and 16% identified as a community member. With respect to overall satisfaction with the conference, the mean response was 4.61 (range 4–5) out of a possible total score of 5, suggesting that delegates were satisfied to very satisfied with the conference. When asked whether they would attend the CSRS in the future, 76% indicated *yes*, 23% indicated *maybe*, and 0% indicated *no*. When asked how often the CSRS should be offered (every year or every two years), 58% indicated every year and 41% indicated every two years. When asked whether future conferences should be offered virtually, in-person, or offered as a hybrid of both in-person and virtual, 59% indicated a preference for a hybrid platform, 29% preferred a fully virtual platform, and 12% indicated preference for an in-person conference. Finally, respondents were invited to submit additional comments, all of which were positive. One delegate shared: “Conference program presented innovative current stress research. Well organized and sufficient breaks throughout the conference. ThinkLink Graphics is very unique and engaging way to present key highlights from a session. Thank you for an enjoyable learning experience at 2021 CSRS”. Another conference delegate shared: “Great experience for me and loved that this was a relatively small conference tailored to my specific interests. Absolutely loved that you included an Indigenous speaker to talk about traditional knowledge, was the highlight of my conference experience so far!”

Although this represents a relatively small sample, responses will inform future planning of the CSRS with respect to flexibility surrounding in-person conference attendance by offering a hybrid platform, the frequency of conference offerings, and the inclusion of creative knowledge dissemination activities. 

## 6. Conclusions

The inaugural Canadian Stress Research Summit was the first conference of its kind to disseminate stress research through a Canadian lens while bridging science and community. This conference highlighted important methodological considerations for understanding the relationship between stress exposure and various outcomes of interest that pertain to the mental health and wellbeing of Canadians. Variability in stress exposure may occur at different life stages, which may have a cumulative effect on health outcomes over time. An examination using a sex and gendered lens, as well as intersections with ethnocultural identities, is paramount to understanding nuances in the data. While the COVID-19 pandemic provided researchers with a unique opportunity to investigate a shared exposure across Canada, individual difference factors of risk and resilience continue to explain the variance in measured outcomes. Strength-based approaches to enhancing stress resilience were highlighted by researchers as well as novel clinical approaches to treating persons with stress-related mental health disorders. What was also clear from the impressive wealth of knowledge shared during this 3-day conference was the need for continued work to understand stress across the lifespan, from an inclusive and diverse Canadian lens.

## Figures and Tables

**Table 1 ijerph-19-11015-t001:** Overview of the conference program.

Thursday, 6 May 2021	
9:30–10:00	Opening Remarks and Welcome
10:00–11:00	Opening Keynote: From Neurotoxicity to Vulnerability: A Developmental Perspective of the Effects of Stress on the Brain
11:00–11:15	Self-Care Break
11:15–12:30	Symposium: Stress, Sex, and Gender
12:30–14:30	Lunch and Virtual Networking
14:30–15:45	Symposium: Stress and Indigenous Health—Teachings of Blood Memory and Epigenetics
15:45–16:00	Self-Care Break
16:00–15:15	Symposium: Stress on the Job
17:30–18:30	Poster Session and Virtual Networking
Friday, 7 May 2021	
9:30–10:00	Opening Remarks and Housekeeping
10:00–11:00	Keynote: A Tale of Translation: Endocannabinoid Regulation of Stress, Anxiety, and Fear, From Rodent to Humans
11:00–11:15	Self-Care Break
11:15–12:30	Symposium: Stress Biomarkers and Mechanisms
12:30–13:30	Lunch and Virtual Networking
13:30–14:45	Symposium: Stress Interventions and Resilience
14:45–15:00	Self-Care Break
15:00–16:15	Symposium: Stress and Development
16:15–16:30	Self-Care Break
16:30–17:45	Symposium: Stress and Clinical Outcomes
Saturday, 8 May 2021	
9:30–10:00	Opening Remarks and Housekeeping
10:00–11:00	General Public Presentation: Mental Health in 2021—The Echo Pandemic
11:00–11:15	Self-Care Break
11:15–12:30	Symposium: The COVID-19 Pandemic
12:30–13:30	Lunch and Virtual Networking
13:30–14:55	Student Spotlight: 5 Minute Flash Talks
14:55–15:05	Self-Care Break
15:05–16:15	Parallel Workshops
16:15–16:30	Self-Care Break
16:30–17:45	Closing Remarks and Awards

## Data Availability

Not applicable.

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
