# Peer review of "Stress across the Lifespan: From Risk to Management—Conference Report on the Inaugural Canadian Stress Research Summit"

_ijerph, 2022, doi:10.3390/ijerph191711015_

Round 1
Reviewer 1 Report
The present manuscript summarizes the activities and themes of the first Canadian Stress Research Summit. Overall, the report is clear, informative and well-written. I have two suggestions where I thought the reader might benefit from additional contextualized information. However, considering the format of the publication, this may be out of the scope of a report, please address my comments as you see fit.
- As one of the many scholars conducting stress research in Canada, I have not come across many conferences dedicated to stress research outside Canada. Many seminal Canadian scholars have forged this field and changed how we understand and treat stress (e.g. Hans Selye, Vivian M. Rakoff, Micheal Meaney, Sonia Lupien). It might be worth adding a few sentences in the introduction about the uniqueness/richness of the Canadian stress research history.
- I appreciate the presentation of the program in the conference structure, but this section felt a little too descriptive. One element that could be added is a bit more information on the intentions of the program committee concerning decisions between choosing certain activities (e.g. live graphical synthesis) and the structure to achieve the conference objectives. It would be interesting for the reader to learn more about how this lens was defined. Finally, one small detail, both the scientific and the conference committee, was used. Are these the same or different committees?
- Finally, I noticed a few typos in lines 80,96,157.
Author Response
Response to Reviewer 1:
I would like to thank Reviewer 1 for their time and consideration in reviewing the submitted manuscript. My response to each comment is itemized below:
The present manuscript summarizes the activities and themes of the first Canadian Stress Research Summit. Overall, the report is clear, informative and well-written. I have two suggestions where I thought the reader might benefit from additional contextualized information. However, considering the format of the publication, this may be out of the scope of a report, please address my comments as you see fit.
Response: I would like to thank the reviewer for their positive overview of the manuscript.
- As one of the many scholars conducting stress research in Canada, I have not come across many conferences dedicated to stress research outside Canada. Many seminal Canadian scholars have forged this field and changed how we understand and treat stress (e.g. Hans Selye, Vivian M. Rakoff, Micheal Meaney, Sonia Lupien). It might be worth adding a few sentences in the introduction about the uniqueness/richness of the Canadian stress research history.
Response: I greatly appreciate the reviewer’s suggestion. For the sake of brevity, and to ensure that the paper remains focused, I have inserted a short paragraph that refers to Canada’s rich history of stress research (first paragraph under Conference Inception and Intentions).
- I appreciate the presentation of the program in the conference structure, but this section felt a little too descriptive. One element that could be added is a bit more information on the intentions of the program committee concerning decisions between choosing certain activities (e.g. live graphical synthesis) and the structure to achieve the conference objectives. It would be interesting for the reader to learn more about how this lens was defined. Finally, one small detail, both the scientific and the conference committee, was used. Are these the same or different committees?
Response: I thank the reviewer for this comment. The new subsection in the introduction titled “Conference Inception and Intentions” provides additional information on development of the conference.
- Finally, I noticed a few typos in lines 80,96,157.
Response: The document was scanned for typos.
Reviewer 2 Report
Dear Author
Thank you for the opportunity to read the manuscript, which I read with great interest.
It is a very good conference report that shows the main concern of organising and disseminating what is being done on stress management in the various stages of life. This report is well structured and very clear.
Only one remark, very simple:
Abstract - The abstract should have the objective of the manuscript and main conclusions. It should be revised.
Section 1 (Introduction) - this section needs some adjustments, as some information and/or points are missing or unclear, and should be included or better written, I will present some items:
- What is the importance of the Summit/contribution it has from the area?
-Why should readers be interested?
-What problem/issue does this conference/report solve/fulfill?
- How will the proposed manuscript remedy this deficiency/lacuna/problem and provide a unique contribution to the scientific community.
Section 3 (Conference Sessions) - no reference to supplementary figure 6 appears in the text.
Section 4 (Conference Analysis and Feedback) - it seems to me that this section could be critically reflective of the results of the speakers' analysis and in that sense could be improved.
Author Response
Response to Reviewer 2:
I would like to thank Reviewer 2 for their time and consideration in reviewing the submitted manuscript. Please find my itemized response to each comment below:
Thank you for the opportunity to read the manuscript, which I read with great interest. It is a very good conference report that shows the main concern of organising and disseminating what is being done on stress management in the various stages of life. This report is well structured and very clear.
Response: I would like to thank the reviewer for their positive overview of the manuscript.
Only one remark, very simple: Abstract - The abstract should have the objective of the manuscript and main conclusions. It should be revised.
Response: The revised abstract includes the objective and conclusions of the conference report.
Section 1 (Introduction) - this section needs some adjustments, as some information and/or points are missing or unclear, and should be included or better written, I will present some items: What is the importance of the Summit/contribution it has from the area? Why should readers be interested? What problem/issue does this conference/report solve/fulfill? How will the proposed manuscript remedy this deficiency/lacuna/problem and provide a unique contribution to the scientific community.
Response: The revised introduction highlights the lack of conference programming available to scholars in the field of stress research despite the rich history of stress research in Canada. This is why the CSRS is unique and addresses a need within the scientific community.
Section 3 (Conference Sessions) - no reference to supplementary figure 6 appears in the text.
Response: I thank the reviewer for noting this omission which has been corrected.
Section 4 (Conference Analysis and Feedback) - it seems to me that this section could be critically reflective of the results of the speakers' analysis and in that sense could be improved.
Response: Given the relatively small % of respondents, the intention is only to provide a descriptive analysis without interpretation. In reflecting on the results, I have included a brief statement on how this information informs future planning of the CSRS.